

# Identifying global patterns of stochasticity and nonlinearity in the Earth System

Fernando Arizmendi[1], Marcelo Barreiro[1], and Cristina Masoller[2]

[1]Instituto de Física, Facultad de Ciencias, Universidad de la República, Iguá 4225, Montevideo, Uruguay
[2]Departament de Física, Universitat Politecnica de Catalunya, 08222 Terrassa, Barcelona, Spain

*Correspondence to:* F. Arizmendi (arizmendi.f@gmail.com)

**Abstract.** By comparing time-series of surface air temperature (SAT, monthly reanalysis data from NCEP CDAS1 and ERA Interim) with respect to the top-of-atmosphere incoming solar radiation (the insolation), we perform a detailed analysis of the SAT response to solar forcing. By computing the entropy of SAT time-series, we also quantify the degree of stochasticity. We find spatial coherent structures which are characterized by high stochasticity and nearly linear response to solar forcing (the shape of SAT time-series closely follows that of the isolation), or vice versa. The entropy analysis also allows to identify geographical regions in which there are significant differences between the NCEP CDAS1 and ERA Interim datasets, which are due to the presence of extreme values in one dataset but not in the other. Therefore, entropy maps are a valuable tool for anomaly detection and model inter-comparisons.

## 1 Introduction

Improving our understanding of the nonlinear and stochastic nature of atmospheric variability is important for advancing long-term forecasts. Climate networks, which are defined over a regular grid of geographical locations (nodes) covering the Earth surface, have proven to be a valuable analysis tool for understanding climate phenomena Tsonis et al. (2006); Tsonis and Swanson (2008); Yamasaki et al. (2008); Donges et al. (2009); Barreiro et al. (2011); Deza et al. (2013); Martin et al. (2013); Tirabassi et al. (2013); Tantet et al. (2013); Rossi et al. (2013); Arizmendi et al. (2014); Feng et al. (2014); Hlinka et al. (2014); Tupikina et al. (2014); Stolvoba et al. (2014); Fountalis et al. (2015); Tirabassi et al. (2015); Martín-Gómez et al. (2015). In climate networks, a link between two nodes is defined in terms of a bivariate statistical similarity analysis of time-series of a climate variable (e.g., surface air temperature, SAT) recorded at each node. However, similar response to solar forcing and/or similar stochastic properties of climate variability can lead to the identification of spurious links which do not represent actual interactions.

The goal of our work is to extract information about local climate properties, by performing univariate time-series analysis. Specifically, we focus on the local response to solar forcing and on the level of stochasticity. By performing a detailed analysis of SAT time-series recorded at a regular grid of nodes, we aim to answer the following questions: where are the regions in which the response to solar forcing is strongly distorted (due to local nonlinear effects and/or nonlocal effects)? and where are





the regions in which SAT variability is more stochastic? In spite of having important implications, to the best of our knowledge,

2   no systematic study in the literature has addressed these issues.

## 2   Datasets

4   We consider monthly mean SAT data from two reanalysis data sets: NCEP CDAS1 Kalnay et al. (1996) and ERA Interim

ERA Interim (2000). The spatial resolution is $2.5^o/1.5^o$ and cover the time-period [1949-2011]/[1979-2013] respectively. The

6   NCEP CDAS1 reanalysis has $N = 10224$ SAT time-series of length $L = 756$ while in ERA Interim, $N = 28562$ of $L = 408$.

The insolation at the top of the atmosphere is calculated following Berger (1978), as a function of day of year and latitude.

8   Then, monthly averaged values for every latitude are calculated.

## 3   Methods of time-series analysis

10   To investigate the local response to solar forcing we compare SAT time-series in each grid point, $y_i(t)$, with the top-of-

atmosphere incoming solar radiation (the insolation), in the same location, $x_i(t)$, assuming an adhoc, heuristic approach that

12   there is a functional relation between them, $y_i = F(x_i)$. We are interesting in assessing the similarity of the shape of $x_i(t)$ and

$y_i(t)$ time-series, which captures the nonlinear nature of the functional relation between them. Therefore, after normalizing the

14   two time-series to have zero mean and unit variance, we calculate the difference between them as,

$$d_i = \sum_{t=1}^{L} |y_i(t) - x_i(t + \tau_i)|. \tag{1}$$

16   Here $\tau_i$ is a shift that takes into account that there can be a lag-time between the two time-series, due, for example, to inertia

and/or memory effects. This lag is expected to be important in the oceans in comparison with land masses, because of the larger

18   heat capacity of water. Thus, at each location, $i$, the lag-time $\tau_i$ is calculated such as to minimize $d_i$. We search the minimum

considering $\tau_i$ values in the interval [0,4] months because of the lack of physical mechanisms that could result in more than

20   four months delay in SAT response with respect to the insolation forcing.

Because we focus on the mean response to solar forcing, $\tau_i$ is calculated to minimize the distance between the isolation and

22   the climatology (the averaged monthly SAT); similar results were found when considering the raw SAT time-series instead.

We note that, if the response is perfectly linear, after normalizing and computing the lag, we have $x_i(t + \tau_i) \sim y_i(t)$ and

24   thus, $d_i \sim 0$. Therefore, by measuring how different in shape the two time-series are, $d_i$ quantifies the distortion in the output

response (SAT time-series) with respect to the input signal (the solar forcing represented by the insolation). It is worth to

26   mention that, in Eq. (1), if instead of using $|y_i(t) - x_i(t + \tau_i)|$ we use $(y_i(t) - x_i(t + \tau_i))^2$, then $d_i$ is equal to $2(1 - \rho_i)$ where

$\rho_i$ is the cross-correlation coefficient between $y_i(t)$ and $x_i(t + \tau_i)$.

28   To quantify the stochasticity of the SAT time-series, we calculate the classical Shannon entropy, defined as

$$H_i = -\sum_n p_n \log p_n / H_{max} \tag{2}$$





where $p_i = \{p_n\}$ with $n = 1 \ldots M$ and $\sum_n p_n = 1$ is the probability distribution associated to the SAT time-series $x_i(t)$,
and $M$ is the number of bins. The entropy is normalized to the maximum value corresponding to the uniform distribution,
$H_{max} = \log M$. $M$ is the same for all time-series within a reanalysis database, but is adjusted in each database to take into
account different length of the time-series: we consider 20 bins for ERA Interim and 40 bins for NCEP CDAS1. This gives
approximately the same number of data points per bin ($\sim 20$).

# 4  Results

Figure 1 presents the results of the forcing-response analysis. Figure 1(a) displays the map of $d_i$ values computed without
shifting the isolation and SAT time-series, Fig. 1(b) displays the same but after shifting them a value $\tau_i \in [0,4]$, and Fig. 1(c)
displays the map of $\tau_i$ values. These maps were obtained from the analysis of the ERA Interim dataset; similar results were
obtained from the NCEP CDAS1 dataset.

In Fig. 1(a) we observe large nonlinear response in the tropics with coherent spatial structures in the oceans, with large $d_i$
values mainly in the cold tongues and areas associated with easterly trades and upwelling processes, while in the continents $d_i$
values are considerably smaller revealing a more linear insolation-SAT relation. There are exceptions, as in the Amazon and
the African rainforest, which show high $d_i$ values.

When the time-series are shifted, Fig. 1(b), the largest $d_i$ values appear over tropical rainforests of Africa and South America.
This can be expected because during the summer rainy season, when the insolation has its highest values, the solar energy
is used for evaporation instead of for heating. In the oceans, also in the tropics, there are coherent spatial structures, with
high $d_i$ values on the equator, which tend to coincide with regions of deep convection in the Atlantic, Pacific and Indian
oceans, including the Intertropical Convergence Zone (ITCZ). Outside the 10S-10N band, the higher $d_i$ values over the eastern
subtropical north Pacific, as compared to the western basin, can be due to the influence of the semipermanent anticyclones and
stratus clouds. High latitude oceans (southern Ocean, Labrador sea, Greenland sea) also show relatively large $d_i$ values, which
can be interpreted as due to the existence of seasonal sea ice in the regions.

The map of $\tau_i$ values, which is shown in Fig. 1(c), uncovers a rather symmetric behavior between both hemispheres, as
expected, having, in general, $\tau_i = 0$ or 1 over continental areas, and $\tau_i = 2$ over the oceans except on the tropical areas where
$\tau_i$ displays values in the [0-4] and in the high latitudes, in sea ice regions, where $\tau_i = 1$. There are two continental regions, in
the African tropical rainforest and in the Amazon rainforest, where $\tau_i = 4$. This can be interpreted as due to the fact that, in
these regions, the annual maximum of precipitation occurs during the summer monsoon.

Figure 2 presents the analysis of the stochasticity of SAT variability. Figures 2(a) and 2(b) display the entropy, $H_i$, computed
from SAT time-series and SAT anomalies (i.e., removing the mean annual cycle from the time series) respectively, using ERA
Interim reanalysis.

The spatial patterns emerging in the ERA Interim SAT entropy map, Fig. 2(a), is in good agreement with the map of $d_i$
values displayed in Fig. 1(b), computed from the same reanalysis after shifting the SAT time-series. We note however that the
values are opposite: regions with high $H_i$ have low $d_i$, and vice-versa.





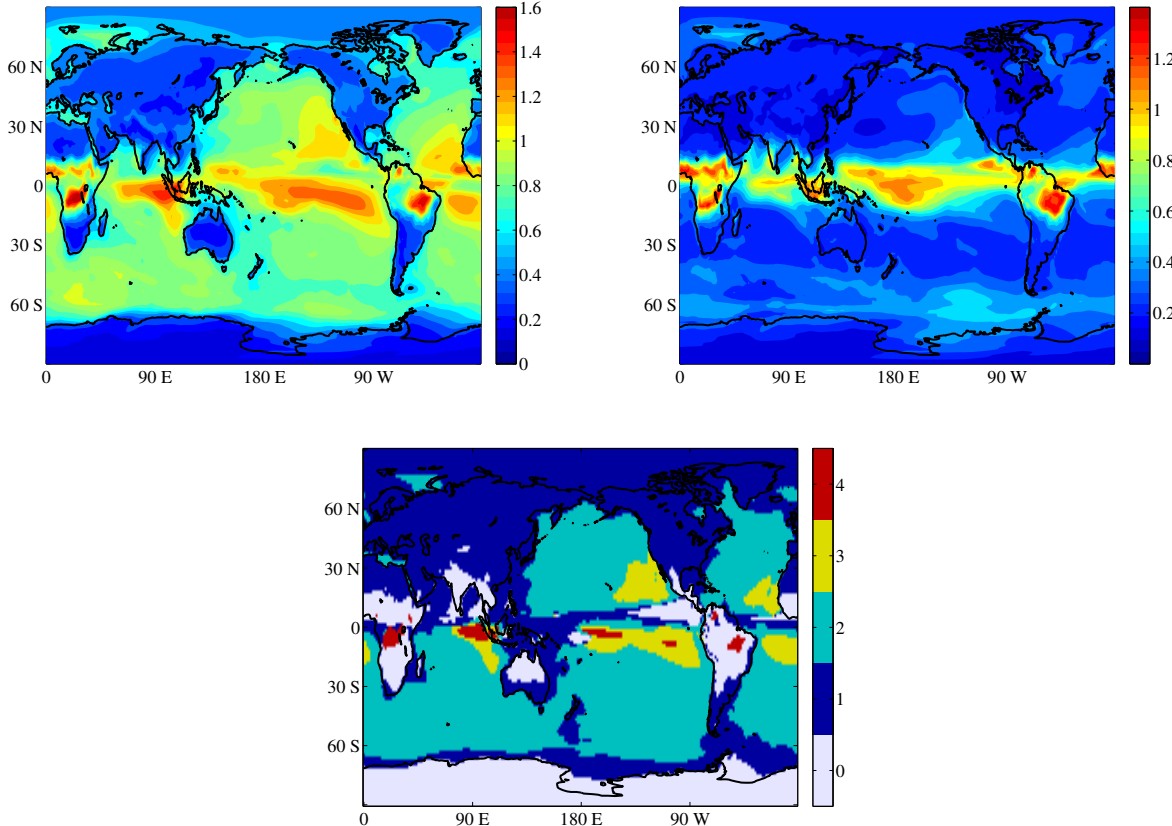

**Figure 1.** (a) Map of distances $d_i$ calculated from Eq. (1) when the forcing (isolation) and the response (SAT) are not shifted ($\tau_i = 0$), (b) Map of distances, when the forcing and the response are shifted $\tau_i$, with $\tau_i \in [0, 4]$. (c) Map of $\tau_i$ values. Data from ERA Interim reanalysis.

In contrast, in the entropy map of SAT anomalies, displayed in Fig. 2(b) also for ERA Interim reanalysis, the main spatial patterns of the tropics seen in Fig. 2(a) disappear, and only those associated with sea ice remain. This might be due to seasonality in the variance, which is not eliminated when removing the annual cycle.

To check whether this is a robust result, we computed the entropy of SAT anomalies using NCEP CDAS 1 reanalysis. The resulting map is presented in Fig. 2(c): it is very similar to that obtained from ERA Interim except in the western tropical Pacific where NCEP data shows minimum entropy. An inspection of SAT time-series in this region reveals the existence of extreme values (outliers) which render the associated probability distribution to be peaked in a narrow interval, which in turn results in low entropy values. In ERA Interim there are also outliers, but, because they cover a very small area, their low entropy values cannot be seen in the entropy map shown in Fig. 2(b). Typical time-series of SAT anomalies in these regions, displaying extreme values, are presented in Fig. 3.





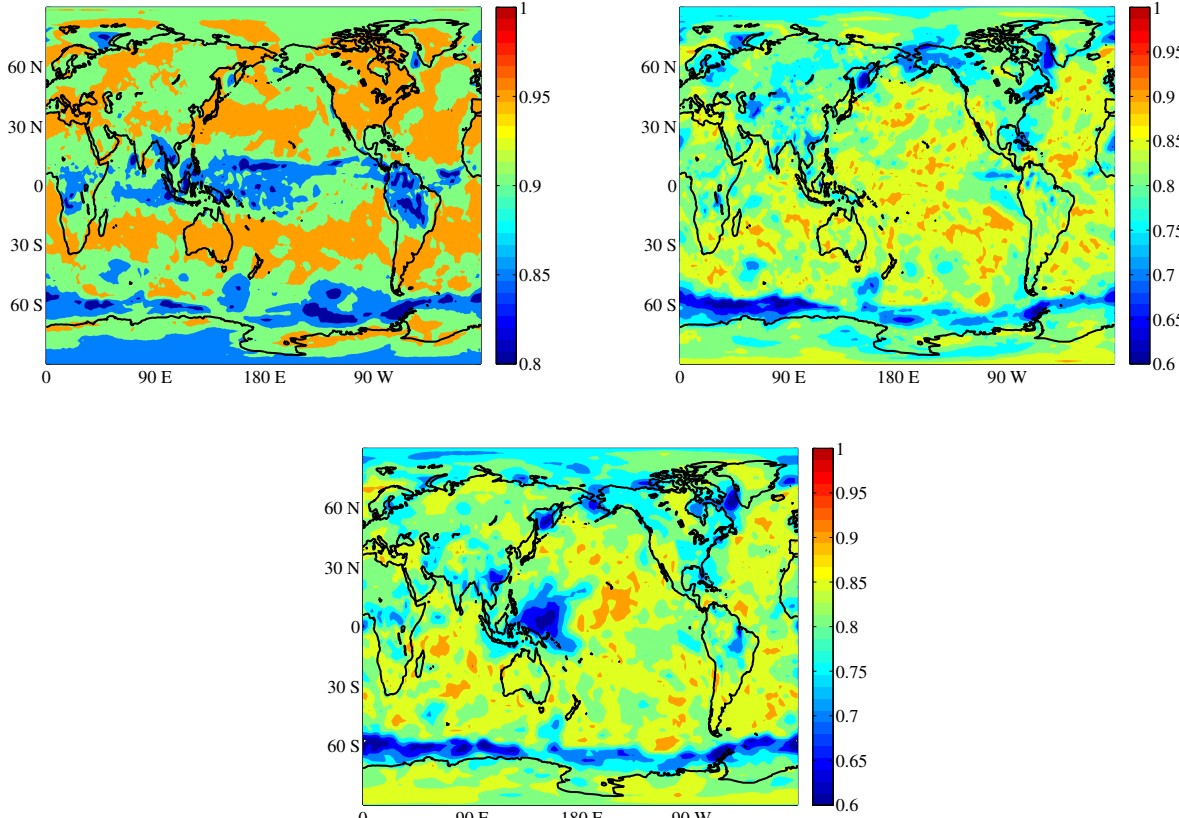

**Figure 2.** Map of Shannon entropy of SAT time-series (a) and of SAT anomalies (b) calculated from NCEP CDAS1 reanalysis. The spatial structures in panel (b) are similar to those observed in Fig. 1 (b). (c) Entropy of SAT anomalies calculated from ERA Interim reanalysis. Comparing panels (b) and (c), we note a well-defined region in the western Pacific where the two data sets display significant differences.

## 5 Conclusions

In this work we have investigated the stochastic properties of a climatological field (the surface air temperature, SAT) and its response to the solar annual forcing. We have characterized SAT stochasticity using Shannon entropy, and we have analyzed the response to solar forcing, by quantifying the similarity in the shape of the time-series of the top-of-atmosphere incoming solar radiation (the insolation) and the SAT, both having been previously normalized to have zero-mean and unit variance. The lag-time between the two time-series was taken into account by lagging the output response (SAT). Two reanalysis datasets were investigated, ERA Interim and NCEP CDAS1.

We found that high shape disimilarity appears mainly in the tropics, with $d_i$ maps [Eq. (1)] displaying well-defined structures in the oceans, over the Intertropical Convergence Zones, and over some continental areas, especially in regions largely dominated by monsoons, such as the tropical rainforest in Africa and south America, as well as over India. The patterns over





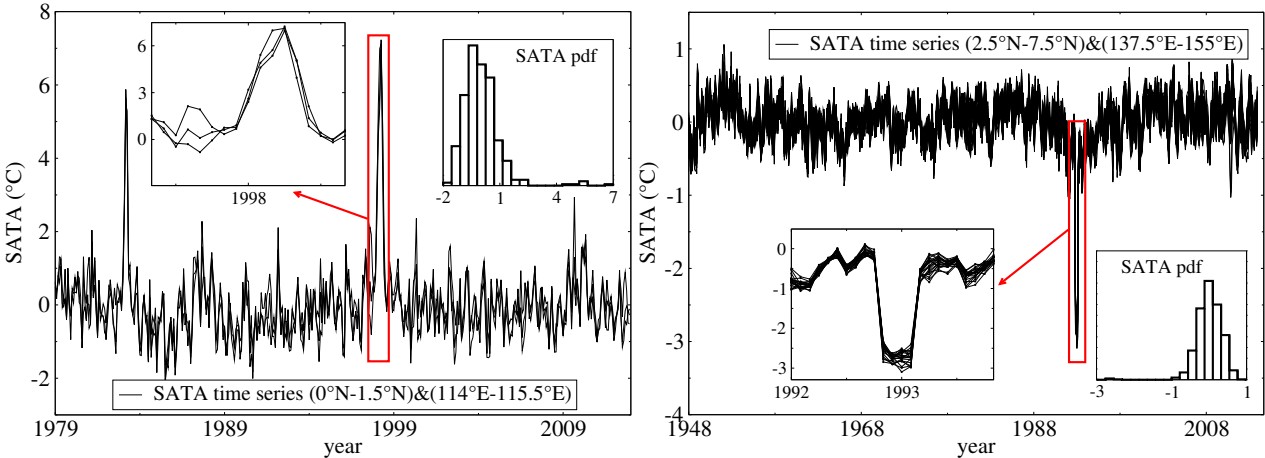

**Figure 3.** SAT anomaly time-series displaying extreme values in NCEP CDAS1 reanalysis (top) and in ERA Interim (bottom). In both cases, the extreme values appear in one dataset but not in the other.

tropical rainforests are expected because during summer, in the rainy season, most of the incoming solar energy is consumed by the water evaporation processes and the surface air temperature remains rather constant.

In the entropy maps we found very similar spatial structures, but with oposite values meaning that the geographical regions with high $d_i$ values correspond, in general, to regions with low entropy values, and vice versa. When the entropy was calculated from the SAT anomalies, the tropical spatial patterns disappeared but those in the high-latitudes remained. This was interpreted as due to the fact that in high latitudes, mainly because of the presence of sea ice, there is a strong seasonality in variance that remains even when the annual cycle is removed.

The entropy analysis also allowed to identify, in a well-defined region of the tropical western Pacific, a remarkable difference between the ERA Interim and the NCEP CDAS1 data sets: in this region the datasets displays extreme values, which cause the pdf to be peaked in a narrow interval, which results in low entropy values. Therefore, entropy maps can be used for anomaly detection and model inter-comparisons.

As future work, it would be interesting to analyze how the global patterns identified here will change under an scenario of climate change.

*Acknowledgements.* This work was supported by the LINC project (FP7-PEOPLE-2011-ITN, Grant No. 289447). C. M. also acknowledges partial support from Spanish MINECO (FIS2015-66503-C3-2-P) and ICREA ACADEMIA.





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
