# Peer review of "Identifying global patterns of stochasticity and nonlinearity in the Earth System"

_Earth System Dynamics, 2016_

## Referee Comment (RC1) · Anonymous Referee #1 · 28 May 2016

First at all, the manuscript contains important formal deficiencies, mainly in the figures and captions, that make difficult to follow it. A partial list of these problems is:

- The figure panels are referenced with letters a), b) . . . but none of the figures have indeed those labels on them.

- In the caption of Fig. 2 the mention to panel b seems to be to panel a instead.

- In the caption of fig 3 top and bottom should be right and left, and the last sentence has no meaning.

- The 'Supplement' file containing Supplementary Information is not mentioned in the text. In caption to fig 1 of Suppl. Information the second 'left' should be 'right', and in the caption of fig 4 left and right seem to be interchanged.

With all these caveats, I am not sure I have understood the paper properly. It is clear that at least these formal aspects should be corrected before resubmission. Nevertheless, assuming I have understood the paper, here are some additional comments on the scientific part:

- Relationship in Eq. (1) is a very particular one. The authors identify deviations from it as 'nonlinearity'. Please note that standard linear relationships such as y(t)=integral_0^t g(t-t') x(t') dt are also linear but different from (1). The authors should comment or state that they are looking at linear relationships involving only the present time and one past time, instead of a more general linear relationship involving a distribution of delays.

- The authors do not state with precision their discretization procedure to define the entropy measure. And the precise form of the discretization is determinant for the entropy values. It is puzzling to read the statement that the presence of outliers 'decreases entropy'. This is exactly the effect contrary to what one should expect from outliers, since the outliers broaden the distribution. Perhaps the authors are adapting the range of values of SATA to the changing range of extreme values at different positions? Only in this way one could expect some entropy decrease with outliers, but this procedure completely destroys any possibility of inter-site comparison. Since there is no statement of neither the range of discretization nor of bin size (only number of bins is stated) there is no way to check the origin of the reported increase of entropy. For fixed bin size and range across the different locations, appearance of extreme values can only increase entropy.

- Page 3: 'this gives approximately the same number of data points per bin': this should apply not to each bin, but to some average value, right?

- The authors attribute the difference in behaviour of the tropical zones to the use of solar radiation in evaporation instead of heating. This sounds reasonable. But one can imagine other possible explanations contributing to this, as for example the smaller amplitude of the variation in solar forcing, or the fact that between the tropics the annual

cycle has two oscillations in solar intensity instead of only one. The authors should comment on that. In general, the identification of the physical phenomena that could be responsible for the observations reported in the paper are rather superficial.

- I found difficult to recognize that the different time series in fig 3 correspond to the different locations inside the stated coordinate range. Perhaps this should be explicitly stated.

---

## Referee Comment (RC2) · Anonymous Referee #2 · 31 May 2016

I reviewed an earlier version of the present manuscript for the journal Physical Review E. For reference, the two reports are attached below. Most of the criticisms raised in those reports still hold and need to be taken into account seriously in a possible major revision of the paper:

* An ad-hoc measure for nonlinear relationships between time series is used, ignoring well established measures that are readily available in the literature. The present results should be compared to those obtained from well-established measures such as mutual information.

* The motivation and aim of the study is not clear: the paper sets of in the introduction with discussing climate networks in the first paragraph. But the rest of the paper does not have anything to do with climate networks.

\* The relationssships between the ad-hoc measure for nonlinear relationships and entropy are discussed only vaguely without supporting statistical analysis. Scatter plots between both quantities would help to support the claims made in the text. Further analysis should follow to substantiate them.

What has improved to some degree in my opinion is the discussion of the physical mechanisms possibly underlying the observed patterns.

—

First review for Physical Review E: ——————————————————

In their article, Arizmendi et al. present their results obtained by applying methods of uni- and bivariate time series analysis to climatological data. Their focus is on quantifying the nonlinearity of the relationship between solar forcing and surface air temperature (SAT) time series as well as the stochasticity of SAT time series. For measuring stochasticity, the well established Shannon entropy is computed based on the time series' empirically estimated probability density function. For measuring the "nonlinearity" of the relationship between two time series, the authors propose a "nonlinear distance" measure (Eq. 1) that only vanishes if the (phase-shifted) time series are identical. Hence, all non-identical (modulo phase shifts) pairs of time series are effectively counted as nonlinearly related. Even for linearly related time series $y(t)= ax(t) + b$, the "nonlinear distance" would be non-zero. For these reasons, I regard this measure as clearly flawed, putting into question some of the main conclusions drawn in the paper concerning "nonlinearity".

The paper contains only minor (if any) methodological advancements and its scope is limited to the field of climatology. The methodological approach and results are not put into a wider context in the paper. This lack of general scope and wider applicability puts into question the paper's relevance for the wider physics readership of PRE. However, pending methodological improvements (see below), the paper contains some results that might be of interest to a climate science audience. Hence, I do not regard the

paper to be suitable for publication in PRE, but suggest that it could be resubmitted to a more specialised climatological journal.

Further comments:

* Regarding the quantification of nonlinear relations between time series, more established and theoretically well-founded methods should be applied, e.g., linear and nonlinear regressions, information-theoretic or recurrence-based approaches etc.

* Starting in Section III, similarities between spatial patterns (maps of scalar measures defined on grid points such as entropy) are discussed a lot in a purely qualitative way. Here it would be advisable to use quantitative means for comparing such patterns (e.g., pattern correlation, scatter plots, mutual information, etc.) to foster a more substantial discussion and conclusions on the relationships between measures and data sets.

—

Second review of revised manuscript for Physical Review E: ————————————— ————————————————

I appreciate the effort invested by the authors in revising their paper. However, I did not find their response to the reviewers' comments persuasive. I still have two main criticisms:

1. The authors introduce an ad-hoc measure of nonlinear relationship between time series, but neither a comparison to established methods for this purpose such as mutual information (or the plethora of other measures in the literature, e.g., Reshef et al., 2011) is given nor is a good motivation for choosing this particular measure other than that it is "simple". Putting the method into context in this way is essential and cannot be considered to be "out of scope" for the paper, particularly since PRE offers sufficient page space to report on such extensive studies.

2. I miss a convincing and well-founded discussion of the physical mechanisms underlying the observed results. The paper could potentially profit much from a detailed

review by an expert in climate physics.

References:

Reshef, D. N., Reshef, Y. A., Finucane, H. K., Grossman, S. R., McVean, G., Turnbaugh, P. J., ... & Sabeti, P. C. (2011). Detecting novel associations in large data sets. science, 334(6062), 1518-1524.

---

## Referee Comment (RC3) · Anonymous Referee #3 · 2 Jun 2016

This manuscripts could be helpful for researchers to decide whether to use linear or nonlinear measures when constructing climate networks. While the material seems to be suitable to the scope of the journal, there are a number of points that I would like to see addressed before the paper is accepted.

I am unclear why they choose $d_i$ as their measure of nonlinearity. They have not referenced where it has been used before, but I get the impression that it is not being introduced here. There are also a large number of techniques for measuring nonlinearity, so a comparison with established methods would also be helpful. Additionally, nonlinearity in the atmosphere has been extensively studied, so a review of the pertinent material should be included in the introduction.

I'm unsure why they believe the large time lag over the Amazon rainforest is due to the

summer monsoon, and I would like them to more fully explain their reasoning.

The strong peak and low entropy values discussed are most likely a result of the binning procedure used, and a more thorough discussion of it is required.

On L:10,11 They define $y_i(t)$ to be the SAT, and $x_i(t)$ to be the solar insolation. However, Eq.1 and their choice of lags suggest that they are searching for the effect of the SAT on the solar insolation at a later time? This does not seem reasonable, and I believe they have included the lag in the wrong term in Eq.1.

As mentioned by other referees, the figures are improperly labelled and captioned, disagreeing with the text at some points.

Isolation is frequently mistakenly used where insolation is meant.

—————————————————

---

## Author Comment (AC1) · 8 Jul 2016

**Responses to reviewers**

We thank the reviewers for a careful reading of our manuscript. In this text, we present the answers to the reviewers comments. Also, we are including below the revised version of the manuscript (modifications in red).

**Response to Reviewer 1**

We thank the reviewer for a careful reading of our manuscript and for listing the mistakes/typos in figure captions etc., which have been corrected in the revised version.

Regarding the additional comments on the scientific part, the reviewer says

"- Relationship in Eq. (1) is a very particular one. The authors identify deviations from it as 'nonlinearity'. Please note that standard linear relationships such as  $y(t)=integral_0^t g(t-t') x(t')$  dt are also linear but different from (1). The authors should comment or state that they are looking at linear relationships involving only the present time and one past time, instead of a more general linear relationship involving a distribution of delays."

Authors' response: we agree with the reviewer and in the revised manuscript we have included the following sentence: We note that more generic relations such as  $y(t)=integral_0^t g(t-t') x(t')$  dt are also linear, however, here we only consider linear relationships involving only the present and only one past time.

**The reviewer says**

"- The authors do not state with precision their discretization procedure to define the entropy measure. And the precise form of the discretization is determinant for the entropy values. It is puzzling to read the statement that the presence of outliers 'decreases entropy'. This is exactly the effect contrary to what one should expect from outliers, since the outliers broaden the distribution. Perhaps the authors are adapting the range of values of SATA to the changing range of extreme values at different positions? Only in this way one could expect some entropy decrease with outliers, but this procedure completely destroys any possibility of inter-site comparison. Since there is no statement of neither the range of discretization nor of bin size (only number of bins is stated) there is no way to check the origin of the reported increase of entropy. For fixed bin size and range across the different locations, appearance of extreme values can only increase entropy."

Authors' response: We disagree with the reviewer that "this procedure completely destroys any possibility of inter-site comparison", on the contrary, we show that in this way the entropy analysis uncovers coherent spatial structures and identifies localized regions where extreme values (likely to be artefacts) are present in the time-series.

These local extreme fluctuations cannot be detected if a fixed bin size and range are used, to compute the entropy, across all datasets. As we say in the manuscript, each time series is first normalized to zero mean and unit variance, and some fluctuations are extreme with respect to the local statistical distribution of SAT (or SATA) fluctuations.

**The reviewer says**

- Page 3: 'this gives approximately the same number of data points per bin': this should apply not to each bin, but to some average value, right?"

Authors' response: We agree with the reviewer and in the revised manuscript we have replaced approximately by average.

**The reviewer says**

- The authors attribute the difference in behavior of the tropical zones to the use of solar radiation in evaporation instead of heating. This sounds reasonable. But one can imagine other possible explanations contributing to this, as for example the smaller amplitude of the variation in solar forcing, or the fact that between the tropics the annual cycle has two oscillations in solar intensity instead of only one. The authors should comment on that. In general, the identification of the physical phenomena that could be responsible for the observations reported in the paper are rather superficial.

Authors' response: We agree that it may additionally have to do with the fact that these are regions with a small amplitude of the seasonal cycle. On the other hand, the insolation we are using to compare with the SAT response has the semi-annual cycle included and if the response were linear the SAT would also have two peaks and the distance would be small.

In the revised version we have also improved the discussion about the physics responsible for the nonlinear response.

**The reviewer says**

- I found difficult to recognize that the different time series in fig 3 correspond to the different locations inside the stated coordinate range. Perhaps this should be explicitly stated.

Authors' response: We have improved the caption of this figure to clarify the panels.

**Response to Reviewer 2**

**The reviewer says**

"An ad-hoc measure for nonlinear relationships between time series is used, ignoring well established measures that are readily available in the literature. The present results should be compared to those obtained from well-established measures such as mutual information."

Authors' response: The measure we use is a known distance between two time-series and in the revised version we have included a sentence about this point including references. In the Supplementary Information we compare with another distance measure and we obtain similar results (Fig. 2 of the SI). The mutual information is not appropriate because the insolation is a fully deterministic periodic signal.

**The reviewer says**

"The motivation and aim of the study is not clear: the paper sets of in the introduction with discussing climate networks in the first paragraph. But the rest of the paper does not have anything to do with climate networks."

Authors' response: We have rewritten the introduction to better motivate this study.

**The reviewer says**

"The relations-ships between the ad-hoc measure for nonlinear relationships and entropy are discussed only vaguely without supporting statistical analysis. Scatter plots between both quantities would help to support the claims made in the text. Further analysis should follow to substantiate them. What has improved to some degree in my opinion is the discussion of the physical mechanisms possibly underlying the observed patterns."

Authors' response: In the revised manuscript we include and discuss the scatter plots. We are pleased that the reviewer considers that the discussion has improved with respect to previous version and we hope that the revised version will convince the reviewer that the analysis performed here uncovered meaningful information about our climate.

**Response to Reviewer 3**

The reviewer says:

"This manuscripts could be helpful for researchers to decide whether to use linear or nonlinear measures when constructing climate networks."

Authors' response: We are pleased that the reviewer appreciates the connection and believes that our work could be helpful for research in climate networks.

The reviewer says:

"While the material seems to be suitable to the scope of the journal, there are a number of points that I would like to see addressed before the paper is accepted.

I am unclear why they choose di as their measure of nonlinearity. They have not referenced where it has been used before, but I get the impression that it is not being introduced here."

Authors' response: We use a common measure of distance between timeseries, and in the revised version we have included a sentence and relevant references.

The reviewer says:

"There are also a large number of techniques for measuring nonlinearity, so a comparison with established methods would also be helpful."

Authors' response: We agree with the reviewer and we remark that in the Supplementary Information we compare with an alternative measure and we obtain similar results (Fig. 2 in the SI).

The reviewer says:

"Additionally, nonlinearity in the atmosphere has been extensively studied, so a review of the pertinent material should be included in the introduction."

Authors' response: In the revised manuscript we have re-written the introduction and include a discussion of previous studies of nonlinearity in the atmosphere.

The reviewer says:

*"I'm unsure why they believe the large time lag over the Amazon rainforest is due to the summer monsoon, and I would like them to more fully explain their reasoning."*

Authors' response: As we say in the revised manuscript, the large lag in the Amazon rainforest does not have a straightforward explanation and we only speculate that a possible reason is due to the summer monsoon. We have modified the manuscript to explain why we consider this is a possible reason.

**The reviewer says:**

"The strong peak and low entropy values discussed are most likely a result of the binning procedure used, and a more thorough discussion of it is required." Authors' response: We agree with the reviewer and in the revised manuscript we clarify this point. We have included this sentence: By normalizing each time series to zero mean and unit variance, we effectively adapt the range of values of SAT/SATA to the changing range of extreme values at different positions. While this might seem contradictory with performing "inter-site comparisons", this allows focusing the analysis in local extreme values (in other words, values that are extreme for the local SAT/SATA distribution).

**The reviewer says:**

"On L:10,11 They define yi(t) to be the SAT, and xi(t) to be the solar insolation. However, Eq.1 and their choice of lags suggest that they are searching for the effect of the SAT on the solar insolation at a later time? This does not seem reasonable, and I believe they have included the lag in the wrong term in Eq.1."

Authors' response: We agree with the reviewer and in the revised manuscript we corrected this mistake.

**The reviewer says:**

"As mentioned by other referees, the figures are improperly labelled and captioned, disagreeing with the text at some points. Isolation is frequently mistakenly used where insolation is meant."

Authors' response: We agree with the reviewer, we have carefully revised the manuscript and corrected these and other mistakes.

**Identifying global patterns of stochasticity and nonlinearity in the Earth System**

Fernando Arizmendi1, Marcelo Barreiro1, and Cristina Masoller2

1Instituto de Física, Facultad de Ciencias, Universidad de la República, Iguá 4225, Montevideo, Uruguay 2Departament de Física, Universitat Politecnica de Catalunya, 08222 Terrassa, Barcelona, Spain *Correspondence to:* F. Arizmendi (arizmendi.f@gmail.com)

**Abstract.** By comparing time-series of surface air temperature (SAT, monthly reanalysis data from NCEP CDAS1 and ERA Interim) with respect to the top-of-atmosphere incoming solar radiation (the insolation), we perform a detailed analysis of the

- SAT response to solar forcing. By computing the entropy of SAT time-series, we also quantify the degree of stochasticity. We
- 4 find spatial coherent structures which are characterized by high stochasticity and nearly linear response to solar forcing (the shape of SAT time-series closely follows that of the insolation), or vice versa. The entropy analysis also allows to identify
- 6 geographical regions in which there are significant differences between the NCEP CDAS1 and ERA Interim datasets, which are due to the presence of extreme values in one dataset but not in the other. Therefore, entropy maps are a valuable tool for
- 8 anomaly detection and model inter-comparisons.

**1 Introduction**

2

- 10 The complex behavior of the atmosphere has attracted large interest in the last decades, as improving our understanding of its nature is important for advancing long term forecasts (e.g., Sugihara et al., 1999). Atmospheric nonlinearity has been studied
- 12 using simplified models (Lorenz, 1963; Kondrashov et al., 2011), analyzing dynamical fields using statistical modal decomposition (Branstator and Berner, 2005) and Lyapunov exponents (e.g., Selten, 1993). Studies have shown that tropical and
- 14 extratropical regions behave very differently as they respond to insolation in disparate manners. In the extratropics hydrodynamical instabilities dominate and the atmospheric behavior is highly nonlinear. For example, baroclinic instability is thought to
- 16 be responsible for the development of cyclones, anticyclones and fronts, that is, the elements of day-to-day weather (Charney, 1947; Trenberth, 1991). These phenomena are very efficient in transporting heat from the tropics to the poles thus decreasing
- 18 the equator-to-pole temperature gradient. Averaging over time scales longer than synoptic, it is thus expected for stochasticity to be large in the extratropics. In the tropics the excess of heat is transported to the extratropics through a thermally direct cell,
- 20 the Hadley circulation. Moreover, on synoptic time scales, while tropical small scale and short-living phenomena are inherently nonlinear, such as convection and the development of mesoscale complexes, the large scale atmospheric variability is primarily
- 22 linear in terms of waves that result from sea and land-surface forcing (Matsuno, 1966; Gill, 1980, 1982).

On the other hand, coupling between the different components of the climate system is strong in the tropical regions. Two 24 well known examples are the monsoons and the ocean cold tongue dynamics. The monsoons are the result of the movement of

1

tropical convection due to seasonal changes in the land-sea temperature contrast and thus do not depend on the local insolation.

- 2 These are regions where the large scale circulation, soil moisture and surface temperature interact strongly generating large climate variations. For example, monsoons have been shown to present nonlinear behavior on intraseasonal time scales (e.g.,
- 4 Chattopadhyay et al., 2008) as well as large interannual variability (e.g., Walker et al., 2015). Similarly, in the equatorial oceans, dynamical air-sea interactions are strong and have been shown to modify the sea surface temperature response to solar
- 6 forcing. For example, while in the eastern equatorial Pacific the insolation presents two maxima as the sun crosses the Equator twice a year, the seasonal cycle of sea surface temperature has only one peak. Several authors have demonstrated that complex
- 8 coupled atmosphere-ocean processes are responsible (e.g., Mitchell and Wallace, 1992; Chang and Philander, 1994). This has also consequences for the latitudinal movement of the Intertropical Convergence Zone: while over land the ITCZ tends to
- 10 follow the sun, over the eastern basins the ITCZ are located almost year round to the north of the equator. Thus, these are all regions where a simple response to the direct solar forcing will not expected to be found.
- 12 The goal of this study is to perform a systematic study of nonlinearity and stochasticity in the atmosphere, which, to the best of our knowledge has not yet been done. We aim to answer the following questions: where are the global regions in
- 14 which the response to solar forcing is strongly distorted due to local or nonlocal nonlinear processes and, which are the regions with largest stochasticity? Is there a relationship between these two quantities? To do so we will compare the local insolation
- 16 with a measure of atmospheric response. We will analyze monthly mean surface air temperature (SAT) because it provides an integrated picture of the Earth system's reponse to insolation as it strongly depends not only on the direct solar forcing but also
- 18 on the transport of heat due to atmospheric processes.

Besides providing a more complete picture of the nature of the atmospheric response to insolation, results may have con-

- 20 secuences for the construction of climate networks. Climate networks, which are defined over a regular grid of geographical locations (nodes) covering the Earth surface, have proven to be a valuable analysis tool for understanding climate phenomena
- 22 (Tsonis et al., 2006; Tsonis and Swanson, 2008; Yamasaki et al., 2008; Donges et al., 2009; Barreiro et al., 2011). In climate networks, a link between two nodes is defined in terms of a bivariate statistical similarity analysis of time-series of a climate
- 24 variable (e.g., SAT) recorded at each node. However, similar response to solar forcing and/or similar stochastic properties of climate variability can lead to the identification of spurious links which do not represent actual interactions.
- Finally, several studies have shown that the annual cycle of surface temperature has changed over the past 60 years toward earlier seasonal phasing and smaller amplitude over most land regions, but a delay in the phase over the ocean (Thomson, 1995;
- 28 Mann and Park, 1996; Wallace and Osborn, 2002; Qian and Zhang, 2015). Mechanisms to explain these changes include loss of sea ice (Dwyer, 2012), large scale decreases in soil moisture which decrease the lag time between insolation and temperature by
- 30 lowering the thermal inertia of the surface (Seager et al., 2007; Stine et al., 2009) and changes in shortwave optical properties that also alter the temperature lag behind insolation (Wallace and Osborn, 2002). Recently, Stine et al. (2012) proposed that changes in the northern extratropics can be related to variations in well known atmospheric circulation patterns. Thus, the
- 2 separation between stochastic regions and those that respond nonlinearly to solar radiation may help in identifying areas that will be sensitive to antropogenic forcing given their complex response to insolation.

**4 2 Datasets**

We consider monthly mean SAT data from two reanalysis data sets: NCEP CDAS1 (Kalnay et al., 1996) and ERA Interim

- 6 (ERA Interim, 2000). The spatial resolution is  $2.5^{\circ}/1.5^{\circ}$  and cover the time-period [1949-2011]/[1979-2013] respectively. The NCEP CDAS1 reanalysis has N = 10224 SAT time-series of length L = 756 while in ERA Interim, N = 28562 of L = 408.
- 8 The insolation at the top of the atmosphere is calculated following Berger (1978), as a function of day of year and latitude. Then, monthly averaged values for every latitude are calculated.

**10 3 Methods of time-series analysis**

To investigate the local response to solar forcing we compare SAT time-series in each grid point,  $y_i(t)$ , with the top-of-12 atmosphere incoming solar radiation (the insolation), in the same location,  $x_i(t)$ , assuming an adhoc, heuristic approach that there is a functional relation between them,  $y_i = F(x_i)$ . We are interesting in assessing the similarity of the shape of  $x_i(t)$  and

- 14  $y_i(t)$  time-series, which captures the nonlinear nature of the functional relation between them. Therefore, after normalizing the two time-series to have zero mean and unit variance, we calculate the difference between them using a rectilinear distance,
- 16 which is known as *taxicab* metric and has been used for example for analyzing urban problems and to assess differences in discrete frequency distributions (Krause, 1975; Lim et al., 2011),

18
$$d_i = \sum_{t=1}^{L} |x_i(t) - y_i(t + \tau_i)|.$$
 (1)

Here τi is a shift that takes into account that there can be a lag-time between the two time-series, due, for example, to inertia
and/or memory effects. This lag is expected to be important in the oceans in comparison with land masses, because of the larger heat capacity of water. Thus, at each location, *i*, the lag-time τi is calculated such as to minimize di. We search the
minimum considering τi values in the interval [0,4] months because of the lack of physical mechanisms that could result in more than four months delay in SAT response with respect to the insolation forcing. Sensitivity to maximum lags is shown in

24 Supplementary Information, Fig. S1.

26

Because we focus on the mean response to solar forcing,  $\tau_i$  is calculated to minimize the distance between the insolation and the climatology (the averaged monthly SAT); similar results were found when considering the raw SAT time-series instead.

- We note that, if the response is perfectly linear, after normalizing and computing the lag, we have  $y_i(t + \tau_i) \sim x_i(t)$  and
- thus,  $d_i \sim 0$ . Therefore, by measuring how different in shape the two time-series are,  $d_i$  quantifies the distortion in the output response (SAT time-series) with respect to the input signal (the solar forcing represented by the insolation). It is worth to
- 30 mention that, in Eq. (1), if instead of using  $|x_i(t) y_i(t + \tau_i)|$  we use  $(x_i(t) y_i(t + \tau_i))^2$ , then  $d_i$  is equal to  $2(1 \rho_i)$  where  $\rho_i$  is the cross-correlation coefficient between  $x_i(t)$  and  $y_i(t + \tau_i)$  (see Fig. S2). We note that more generic relations such as
- 2  $y(t) = \int_0^t g(t t')x(t')dt$  are also linear, however, here we only consider linear relationships involving only the present and only one past time.

4 To quantify the stochasticity of the SAT time-series, we calculate the classical Shannon entropy, defined as

$$H_i = -\sum_n p_n \log p_n / H_{max} \tag{2}$$

- 6 where  $p_i = \{p_n\}$  with n = 1...M and  $\sum_n p_n = 1$  is the probability distribution associated to the SAT time-series  $x_i(t)$ , and M is the number of bins. The entropy is normalized to the maximum value corresponding to the uniform distribution,
- 8  $H_{max} = \log M$ . M is the same for all time-series within a reanalysis database, but is adjusted in each database to take into account different length of the time-series: we consider 20 bins for ERA Interim and 40 bins for NCEP CDAS1. This gives an
- 10 average of the same number of data points per bin ( $\sim$  20). By normalizing each time series to zero mean and unit variance, we effectively adapt the range of values of SAT/SATA to the changing range of extreme values at different positions. While this
- 12 might seem contradictory with performing "inter-site comparisons", this allows focusing the analysis in local extreme values (in other words, values that are extreme for the local SAT/SATA distribution).

**14 4 Results**

Figure 1 presents the results of the forcing-response analysis. Figure 1(a) displays the map of di values computed without
shifting the insolation and SAT time-series, Fig. 1(b) displays the same but after shifting them a value τi ∈ [0,4], and Fig. 1(c) displays the map of τi values. These maps were obtained from the analysis of the ERA Interim dataset; similar results were

- 18 obtained from the NCEP CDAS1 dataset (see Fig. S3).In Fig. 1(a) we observe large nonlinear responses in the tropics with coherent spatial structures over the oceans, with large
- 20  $d_i$  values mainly in the cold tongues and areas associated with easterly trades and upwelling processes. Over the continents  $d_i$  values are considerably smaller revealing a more linear insolation-SAT relation. There are exceptions, as in the Amazon and
- 22 the African rainforest, which show high  $d_i$  values.

When the time-series are shifted, Fig. 1(b), the largest di values appear over the tropical rainforests of Africa and South
24 America. These are the regions dominated by monsoons and can be expected because during the summer rainy season, when the insolation has its highest values, the solar energy is used for evaporation instead of for heating. Moreover, as mentioned in

- 26 the introduction, the development of the monsoon depends on the land-sea contrast and not only on the local insolation. In the oceans, also in the tropics, there are coherent spatial structures, with high  $d_i$  values on the equator, which tend to
- 28 coincide with regions of deep convection in the Atlantic, Pacific and Indian oceans, including the Intertropical Convergence Zone (ITCZ). In these regions the SAT and rainfall are strongly coupled so that relatively small changes in SAT gradients
- 30 modulate and shift the ITCZ. In particular, air-sea coupling in the eastern basins induce oceanic cold tongues which together with the continental geometry maintain warm waters and the ITCZ to the north of the equator (Philander et al., 1996) thus
- 32 introducing a nonlinearity in the SAT reponse. Outside the  $10^{\circ}$ S- $10^{\circ}$ N band, the higher  $d_i$  values over the eastern subtropical north Pacific, as compared to the western basin, can be due to the influence of the semi-permanent anticyclone and associated
- 2 stratus clouds. These clouds shield solar radiation and cool the sea surface, which in turn increases the static stability of the atmosphere producing more stratus decks. This positive stratus-surface temperature feedback thus results in a nonlinear

Figure 1. (a) Map of distances  $d_i$  calculated from Eq. (1) when the forcing (insolation) and the response (SAT) are not shifted ( $\tau_i = 0$ ), (b) Map of distances, when the forcing and the response are shifted  $\tau_i$ , with  $\tau_i \in [0, 4]$ . (c) Map of  $\tau_i$  values. We exclude regions from Congo and Amazon rainforest (dark grey colored) where an increase of insolation is not necessarely an increase in SAT (see text). Data from ERA Interim reanalysis.

180 E

90 W

90 E

0

- 4 response to the insolation. High latitude oceans (southern Ocean, Labrador sea, Greenland sea) also show relatively large  $d_i$  values, which can be interpreted as due to the existence of seasonal sea ice in the regions.
- 6 The map of  $\tau_i$  values, which is shown in Fig. 1(c), uncovers a rather symmetric behavior between both hemispheres. Overall, extratropical land masses have a lag of about 1 month, while extratropical oceans present a lag of 2 months, in agreement with
- 8 McKinnon et al. (2013). In tropical oceanic areas  $\tau_i$  displays values in the [0-4] range, with values close to 0 and 1 in the ITCZ region and  $\tau_i = 3$  in the eastern basins dominated by stratus clouds.

There are two continental regions, in the African tropical rainforest and in the Amazon rainforest that behave rather dif-

2 ferently. In these regions, the annual maximum of precipitation occurs during the summer monsoon, that is, precipitation is maximum at the time of maximum insolation.

---

## Editor Comment (EC1) · M. Crucifix (Editor) · 2 Aug 2016

First of all I would like to thank the authors and all the reviewers for their contributions to Earth System Dynamics Discussions.

We have three referee reports. The authors have replied to the referees and went a bit ahead of the process by providing a revised version. It is however probably premature to send this revised version to the editors for reasons that I will outline below.

First of all let us summarise what are the possible good points and potential issues with this manuscript.

The article considers (local) incoming insolation and reanalyses surface air temperature (SAT) over the last 60 and 40 years (two datasets are used) to establish the degree

of 'linearity' and 'stochasticity' in the surface air temperature response to insolation. The description of 'linearity' and 'stochasticity' relies on measures: a lag-distance and the Shannon entropy of the distribution respectively. The reviewers expressed concerns about the relevance of these indicators.

1. the lagged distance will be non-zero for general forms of linear convolutions. This is a serious objection in principle, and the authors may not have fully acknowledged the point in their revised version since the sentence "if the response is perfectly linear [...] $d_i \approx 0$" remains. However, the smooth, harmonic character of the insolation forcing is such that this might be less of a concern in the end and it must also be noted that the authors provided, in their response, further sensitivity tests using a correlation distance metric.

2. the Shannon entropy has some counter-intuitive behaviours, as for example, a decrease in entropy in presence of extremes. The authors indeed analysed and explained why entropy decreases in their response, but the wider concern as whether entropy is indeed a suitable measure of "stochasticity" remains there. I believe that the authors could be more clear about what is meant here by "stochasticity" here. The very notion of stochasticity is attached to a modelling framework ("someone's noise is someone else's signal") Intuitively, we would be probably see it as a departure from a smooth response to insolation (even though smoothness and predictability are in principle not the same thing) and more arguments must be provided about why entropy could be a good measure of such well-defined stochasticity. Exploring alternative measures of such stochasticity would provide added value to the study.

Reviewer #2 is quite concerned about the overall added value of the paper. I share this concern. We probably need to admit that the physical message is not overwhelming (smaller lag over the continent; non-linearity in the monsoon regions: all this is

pretty expected). Where non-trivial interpretation are mentioned (e.g.: a role for stratus clouds) they are not supported by a full investigation of relevant diagnostics. On the other hand, the methodological implications for climate network theory remain rather oblique. Consider in particular the last two paragraphs of the revised version:

"the entropy analysis also allowed to identify, in a well-defined region of the tropical western Pacific, a remarkable difference between the ERA and the NCEP datastets" [and a discussion about extreme values follows]. This comment seems more to relate to the weakness of the present choice of entropy, and does not relate to climate networks. Then:

"Our results suggest that SAT over the tropical oceanic and continental regions [...] may be the most sensitive to anthropogenic forcing because their evolution depend on a delicate coupling involving air-sea-land processes." This sudden apparition of anthropogenic forcing at the end of the manuscript is rather disconcerting, and it is quite unclear how the results shown here strongly support this statement beyond what is generally admitted.

Regarding the form, the authors extensively edited the first version of the manuscript but these modifications are in need of further editing. Typos, obscure or long sentences remain too numerous. E.g.: "Averaging over time scales longer than synoptic, it is thus expected for stochasticity to be large in the extratropics" or "... is primarily linear in terms of waves that result from sea and land-surface forcing", "We are interesting in assessing...".

As a final note, it is always slightly embarrassing to be informed by a reviewer that a manuscript has already been considered for review in a different journal. This is the reason why "Earth System Dynamics" invites authors to inform the editors of previous submissions, an option that has not been chosen by the authors in the present case.

In conclusion, I would recommend to revise the study (and not only the manuscript) more extensively than proposed by the authors. It will be hard to provide messages of

particular physical significance as long as only SAT is analysed. A suggestion could be to focus more on the implications for network theory, in which case a wider class of measures needs to be analysed and discussed. These recommendations imply a formal decision of "rejection" but the authors are nevertheless encouraged to resubmission.